# Thinned random measures for sparse graphs with overlapping communities

**Federica Zoe Ricci**
Department of Statistics
University of California, Irvine, CA, USA
fzricci@uci.edu

**Michele Guindani**
Department of Biostatistics
University of California, Los Angeles, CA, USA
mguindani@g.ucla.edu

**Erik B. Sudderth**
Departments of Computer Science and Statistics
University of California, Irvine, CA, USA
sudderth@uci.edu

## Abstract

Network models for exchangeable arrays, including most stochastic block models, generate dense graphs with a limited ability to capture many characteristics of real-world social and biological networks. A class of models based on completely random measures like the generalized gamma process (GGP) have recently addressed some of these limitations. We propose a framework for thinning edges from realizations of GGP random graphs that models observed links via nodes' overall propensity to interact, as well as the similarity of node memberships within a large set of latent communities. Our formulation allows us to learn the number of communities from data, and enables efficient Monte Carlo methods that scale linearly with the number of observed edges, and thus (unlike dense block models) sub-quadratically with the number of entities or nodes. We compare to alternative models for both dense and sparse networks, and demonstrate effective recovery of latent community structure for real-world networks with thousands of nodes.

## 1 Introduction

Given observations of (often binary) relationships $Y_{ij}$ between pairs of nodes or entities $(i, j)$, many relational models [1] seek to uncover an underlying set of communities. Classic stochastic blockmodels [2] generalize mixture models for clustering non-relational data by assigning each entity to one of $K$ communities (clusters). The *infinite relational model* (IRM) [3] instead uses a Dirichlet process prior [4] to partition entities into single communities. While the IRM allows the number of communities to be inferred from data, later work has shown that real-world social networks are better captured by models which allow nodes to participate in multiple communities [5], including applications of the *hierarchical Dirichlet process* (HDP) [6] to relational data [7]. Node relationships may also be modeled by shared features [8, 9] learned via the *Indian Buffet Process* [10], by a combination of node and interaction factors [11], or by proximity in a latent space [12, 13].

There is an extensive literature on descriptive statistics of biological and social networks [14, 15] including degree distributions, path distances and "small world" phenomena [16], community structures and modularity, and notions of centrality and causality. In particular, sparsity is a ubiquitous phenomenon in real-world networks [14, 15]: as network size grows, the number of edges grows more slowly than the quadratic number of node pairs. However, the IRM and HDP relational models (and a large literature of related models [1]) generate *dense* graphs where the number of edges scales quadratically with the number of nodes. In fact, a classic representation theorem [17, 18] shows

that any generative model that regards graphs as exchangeable adjacency matrices, meaning arrays whose distribution is invariant to permutations of node indices, generates dense graphs. Most existing probabilistic network models, which generate an adjacency matrix by sampling edges from independent Bernoulli distributions given latent node-specific parameters, have this limitation. Another important property of many real-world networks is assortative mixing [19, 20, 21], that is the presence of more popular or sociable nodes to which other nodes are more likely to connect. Models with degree-correction mechanisms [22, 23, 24] account for this phenomenon.

By representing graphs as a latent process (a completely random measure), Caron and Fox [25] showed that it is possible to formulate generative models that capture the sparsity of real-world networks as well as assortative mixing. Related models, including certain infinite limits of graphs called *graphons*, have been studied by several authors [26, 27]. However, these models mostly produce homogeneous graphs with sparsity and heavy-tailed degree distributions, but lacking the community structure of real networks. Two notable exceptions are work by Herlau et al. [28] and Todeschini et al. [29] that augment random-measure models [25] with latent community structure. Alternative approaches to sparse networks regard edges, rather than nodes, as the core of the generative process [30, 31, 32]. Another approach to simultaneously capture local density and global sparsity models graphs as a collection of cliques [33].

In this paper, we propose a novel random graph model that efficiently thins edges from sparse homogeneous graphs to reveal community structure while maintaining sparsity. Unlike [28], we allow entities to have membership in multiple communities. Unlike [29], we allow completely flexible specification of the hierarchical Bayesian prior on latent community memberships; importantly, this enables the network-specific learning of the appropriate number of latent communities. We further develop an efficient Monte Carlo inference algorithm that, unlike nearly all dense block models, scales linearly with the number of observed edges and thus sub-quadratically with the number of entities. Experiments show recovery of communities for networks with thousands of nodes.

## 2   Background: Stochastic Blockmodels for Dense and Sparse Networks

An undirected binary network with $N$ nodes and $E$ edges may be represented by a symmetric $N \times N$ adjacency matrix $Y$. If there is an edge (link) between nodes $i \neq j$ then $Y_{ij} = 1$, otherwise $Y_{ij} = 0$.

### 2.1   Mixed Membership Stochastic Blockmodels

*Stochastic blockmodels* (SBMs) [2] assume that each node belongs to one of $K$ latent communities, and that the probability of an edge depends on how strongly their communities are connected. The community $c_i \in \{1, \ldots, K\}$ of node $i$ follows a categorical distribution $c_i \overset{\text{ind}}{\sim} \text{Cat}(\beta)$, where $\beta = (\beta_1, \ldots, \beta_K)$ so that $\beta_k$ controls the relative size of community $k$. Edges are sampled independently as $Y_{ij} \overset{\text{ind}}{\sim} \text{Bernoulli}(\eta_{c_i c_j})$, where $\eta_{k\ell} = \eta_{\ell k}$ is the probability of an edge between nodes in communities $k$ and $\ell$. These interaction probabilities are often assigned conjugate beta priors, $\eta_{k\ell} \overset{\text{ind}}{\sim} \text{Beta}(\tau_a, \tau_b)$.

*Mixed membership stochastic blockmodels* (MMSBs) [5] extend SBMs to allow nodes to be members of multiple communities. Let $\pi_i = (\pi_{i1}, \ldots, \pi_{iK})$ denote a $K$-dimensional probability vector representing the strength of affiliation of node $i$ to each of $K$ communities. For every pair of nodes $(i, j)$, the communities governing their interaction are sampled as $c_{ij} \overset{\text{ind}}{\sim} \text{Cat}(\pi_i)$, $c_{ji} \overset{\text{ind}}{\sim} \text{Cat}(\pi_j)$. Then, like standard SBMs, edges are sampled independently as $Y_{ij} \overset{\text{ind}}{\sim} \text{Bernoulli}(\eta_{c_{ij} c_{ji}})$. Community memberships are typically assigned a hierarchical prior, such as

$$\pi_i \mid \beta \overset{\text{ind}}{\sim} \text{Dirichlet}(\zeta\beta_1, \ldots, \zeta\beta_K), \qquad \beta \sim \text{Dirichlet}\left(\frac{\gamma}{K}, \ldots, \frac{\gamma}{K}\right). \qquad (1)$$

The concentration parameter $\zeta$ controls the polarization of $\pi_i$ and its variation around $\beta$, with smaller values of $\zeta$ inducing more polarized community memberships that place significant probability on only a few communities, and larger values of $\zeta$ inducing community memberships that differ very little from $\beta$. When fitting this model to data, we can treat $K$ as an upper bound to the number of communities (that is, larger than the maximum number of communities that we expect to be necessary to model the network), fix $\gamma \ll K$ and let $\zeta$ be small. In this way, the hierarchical Dirichlet formulation allows network-specific learning of the number of communities by favoring sparse

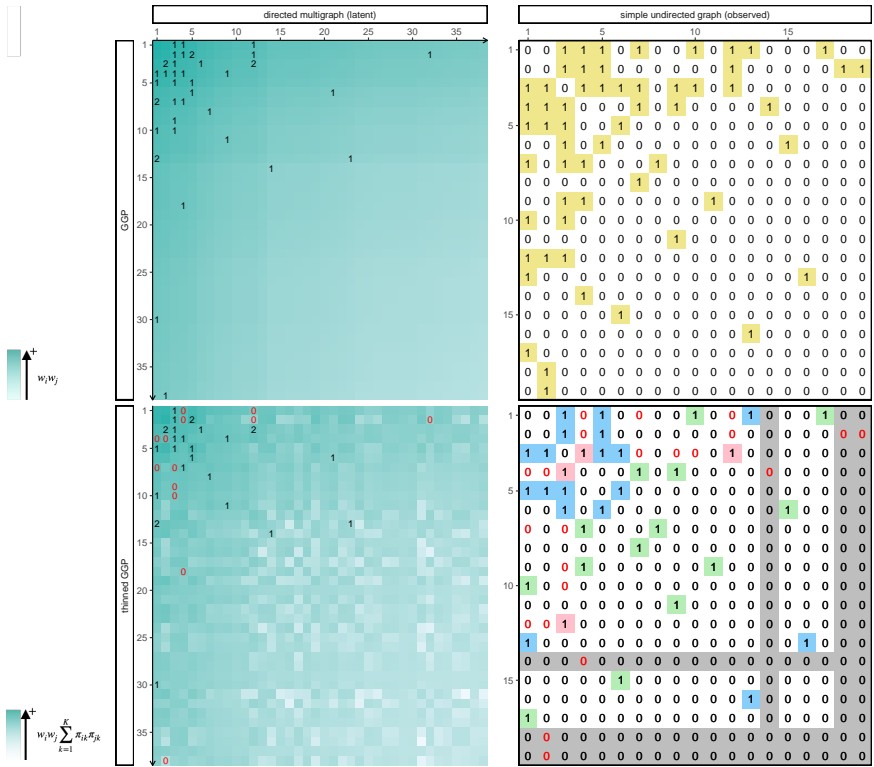

Figure 1: Generation of an adjacency matrix from the GGP model of Caron and Fox [25] (top) and via thinning via our proposed model with $K = 3$ communities (bottom). Rows and columns correspond to nodes in all plots, ordered by decreasing sociability $w_i$. *Left:* Latent directed multigraphs underlying the observed graphs on the right. Darker cells correspond to higher Poisson rates: $w_i w_j$ for the GGP, and $w_i w_j \sum_{k=1}^{K} \pi_{ik} \pi_{jk}$ for the thinned GGP. Red zeros indicate thinned edges. *Right:* Binary adjacency matrix of the observed graph for the GGP (top) and thinned GGP (bottom) models. For the thinned GGP, edges are colored according to the community that generated them. Gray cells mark nodes whose edges in the latent multigraph were all thinned, and are thus observed under the GGP but unobserved under the thinned GGP. GGP hyperparameters were set as $\sigma = 0.1, \tau = 1, \alpha = 10$, and community memberships were sampled given $\beta = (1/3, 1/3, 1/3), \zeta = 1$.

community frequency vectors $\beta = (\beta_1, \dots, \beta_K)$ with some elements $\beta_h \approx 0, h \in \{1, \dots, K\}$ [34]. The hierarchical Dirichlet process [6, 7] is the limit of this prior as $K$ approaches infinity [35].

## 2.2 Sparse Network Models via Completely Random Measures

For any fixed community frequencies $\beta$ and interaction probabilities $\eta$, the SBM and MMSB may only generate *dense* graphs where the number of edges scales quadratically with the number of nodes [18]. In contrast, many real-world networks appear to be sparse [15]. Heuristics are often used to fit (mixed membership) SBMs to large but sparse networks, such as fixing (rather than learning) $\eta_{k\ell} = \varepsilon \approx 0$ for $k \neq \ell$; for example Kim et al. [7] fix $\varepsilon = e^{-30}$. We seek to build models that can capture mixed memberships and sparsity simultaneously without needing to rely on such heuristics.

**Completely Random Measures.** By representing the graph as a point process on the plane, Caron and Fox [25] showed that it is possible to generate sparse graphs by appropriately choosing the mean measure of the point process. According to their model, for $i \neq j$,

$$Y_{ij} \mid w_i, w_j \overset{\text{ind}}{\sim} \text{Bernoulli}\left(1 - \exp\left\{-2 w_i w_j\right\}\right), \tag{2}$$

where $w_i > 0$ represents the *sociability* of node $i$: nodes with higher $w_i$ have higher probability to interact with other nodes, and hence greater expected degree. Differently from the (mixed membership) SBMs, where all parameters governing nodes' interactions are sampled independently, here nodes' sociabilities are generated altogether using the jumps of a *completely random measure* (CRM) [36]. Each node is also independently associated to a real-valued location $\ell_i$ uniformly

distributed on the real line, and then a set of potential nodes is defined by restricting to sociabilities with a sampled location in the interval $[0, \alpha]$,

$$W_\alpha = \{w_i : \ell_i \in [0, \alpha]\}. \tag{3}$$

Here $\alpha > 0$ controls the (random) number of nodes $N_\alpha$ in the network by determining the size of the interval in which jumps are included. Depending on the distribution of the jumps, the model can capture both sparse and dense graphs. Intuitively, as detailed in Sec. 5.1 of [25], for sparse networks the distribution of the jumps needs to place almost all of its mass near zero.

This model requires CRMs for which the sum $\bar{W}_\alpha = \sum_{i:\ell_i \in [0,\alpha]} w_i$ of the jumps in $[0, \alpha]$ is finite. A simple undirected network is then generated via a binary projection of an underlying directed multigraph, where the total number of edges $n_{ij}$ from node $i$ to node $j$ is independently distributed as

$$n_{ij} \mid w_i, w_j \overset{\text{ind}}{\sim} \text{Poisson}(w_i w_j). \tag{4}$$

An edge is present in the undirected network if there is at least one directed edge in the latent directed multigraph, that is $Y_{ij} = \mathbb{1}(n_{ij} + n_{ji} \geq 1)$ for nodes $i \neq j$. Because the sum of independent Poisson random variables is Poisson, Eq. (4) implies $P(n_{ij} + n_{ji} = 0) = \exp\{-2w_i w_j\}$, from which Eq. (2) follows. The number $N$ of nodes in the observed network then equals the number $N_\alpha$ of nodes that have at least one edge in the underlying multigraph. This construction of the binary matrix from a multigraph [25] is visualized in Fig. 1(top).

The sum-property of the Poisson distribution also implies that, given $\bar{W}_\alpha$, the total number of edges $D_\alpha$ in the multigraph has a $\text{Poisson}(\bar{W}_\alpha^2)$ distribution. Since $n_{ij}$ has a high probability of being 0 for most node pairs $(i, j)$, it is more efficient to first sample $D_\alpha$ and then independently assign each edge to a pair of nodes based on their sociabilities. In more detail, Eq. (4) is equivalent to

$$D_\alpha \mid \bar{W}_\alpha \sim \text{Poisson}(\bar{W}_\alpha^2), \quad P(x_{ev} = i \mid W_\alpha) = \frac{w_i}{\bar{W}_\alpha}, \quad n_{ij} = \sum_{e=1}^{D_\alpha} \mathbb{1}(x_{e1} = i)\mathbb{1}(x_{e2} = j), \tag{5}$$

where $x_{ev} \in \{1, 2, \dots\}$ for $v = 1, 2$ indicates the nodes sampled for edges $e \in \{1, \dots, D_\alpha\}$. Caron and Fox [25] propose the *generalized gamma process* (GGP) [37, 38, 39] as a flexible but tractable CRM for $W_\alpha$, with parameters $\tau \in (0, \infty)$, and $\sigma \in (-\infty, 0]$ for dense graphs or $\sigma \in (0, 1)$ for sparse graphs. Fig. 2 (black component) provides a visual summary of the generative process underlying the construction of a GGP random graph, using a directed graphical model that highlights the conditional dependencies (and independencies) between its variables. Fig. 3a shows a network sampled from the GGP model by means of a node-edge diagram.

**Sparse Block Models.** A limitation of the framework described above is that it does not model the community (block) structure of the network – a well-recognized feature of complex networks. Herlau et al. [28] generalized the approach in [25] to accommodate networks with community structure. Specifically, they introduce a latent discrete variable $c_i \in \{1, \dots, K\}$ to indicate the assignment of node $i$ to one of $K$ communities, like in the SBMs. A bivariate CRM incorporates both the sociability weights and a set of parameters, denoted by $\eta_{c_i c_j}$, capturing the interaction strength between two communities $c_i$ and $c_j$ in the underlying multigraph. For their formulation, the total number of edges $n_{ij}$ from node $i$ to node $j$ are independently distributed as $n_{ij} | c_i, c_j, w_i, w_j \overset{\text{ind}}{\sim}$ Poisson($\eta_{c_i c_j} w_i w_j$), and the likelihood from Eq. (2) of the observed nodes is modified as $Y_{ij} \mid c_i, c_j, w_i, w_j \overset{\text{ind}}{\sim}$ Bernoulli($1 - e^{-2\eta_{c_i c_j} w_i w_j}$). We refer to this approach as the *stochastic block model generalized gamma process* (SBM-GGP).

**Sparse Mixed Membership.** The SBM-GGP models sparse and dense networks with community structure, but does not capture overlapping community structures; it induces networks whose nodes are partitioned into disjoint communities. In follow-up work, Todeschini et al. [29] extended the CRM-based framework discussed above by associating a vector $(w_{i1}, \dots, w_{iK})$ of sociabilities to each node $i$, to represent different levels of affiliation of nodes to the $K$ latent communities. A node may have high levels of affiliation to more than a community, leading to the formation of edges across multiple communities. The vectors of node sociabilities are distributed according to a compound CRM [40]; specifically, their implementation uses a *compound generalized gamma process* (CGGP). The likelihood function is modified accordingly, by independently sampling undirected edges as $Y_{ij} \mid w_i, w_j \overset{\text{ind}}{\sim}$ Bernoulli($1 - \exp\{-2\sum_{k=1}^{K} w_{ik} w_{jk}\}$). The latent community weights are modeled

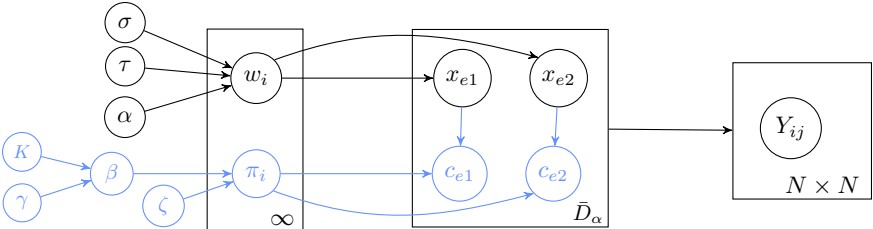

Figure 2: Directed graphical model representing the GGP random graphs of Caron and Fox [25] (black), and the additional variables (blue) in our thinned GGP. For GGP hyperparameters that induce sparse graphs ($\sigma > 0$) there is an infinite number of *potential* nodes, with sociabilities $w_i$ and community memberships $\pi_i$.

as $w_{ik} = w_{i0}\psi_{ik}$, where $w_{i0}$ are sampled from a GGP as in Eq. (3), and $\psi_{ik}$ are gamma-distributed random variables $\psi_{ik} \overset{\text{ind}}{\sim} \text{Gamma}(a_k, b_k)$ whose shape $a_k$ and rate $b_k$ may be inferred from data.

As [29] point out, the CGGP model exploits an (unconstrained) non-negative matrix factorization to define the Bernoulli probability link. Our results in Sec. 5.1 suggest that this lack of constraints affects the ability of the CGGP model to recover the true community structure of the network. Better regularized approaches to non-negative matrix factorization, like the one we propose, may improve the identifiability of latent community structures and network parameters [41].

## 3 The Thinned GGP Network Model

We now add overlapping community memberships to the GGP network model [25] via vectors of probabilities sampled from a hierarchical Dirichlet distribution. Unlike the formulation of Todeschini et al. [29], our model enables learning the number of communities from data (see Secs. 2.1 and 4). By using probability vectors rather than unconstrained non-negative values to model community memberships, our model provides a regularized approach to inference in the GGP model, which in simulations (Sec. 5.1) provides a substantial increase in community detection accuracy.

Let each node $i$ have both a sociability parameter $w_i$ from the GGP as in (3), and a vector of probabilities $\pi_i = (\pi_{i1}, \ldots, \pi_{iK})$ drawn from a hierarchical Dirichlet distribution as in (1). Moreover, let the number of *potential* edges between nodes $i$ and $j$ in the latent multigraph depend only on their sociabilities, as in (4). For each of the $n_{ij}$ potential edges, node $i$ independently samples a community from $\text{Cat}(\pi_i)$, and node $j$ samples a community from $\text{Cat}(\pi_j)$. If these community assignments match, the edge is retained; otherwise, it is *thinned* (i.e., discarded). See Figs. 1 and 3 for examples. More formally, let $\dot{n}_{ij}$ be the number of multigraph edges between a pair of nodes $i, j$ that is retained (not thinned). Edges $e \in \{1, \ldots, D_\alpha\}$ in the GGP multigraph are stochastically thinned as follows:

$$c_{e1} \mid x_{e1}, (\pi_1, \pi_2, \ldots) \overset{\text{ind}}{\sim} \text{Cat}(\pi_{x_{e1}}), \qquad c_{e2} \mid x_{e2}, (\pi_1, \pi_2, \ldots) \overset{\text{ind}}{\sim} \text{Cat}(\pi_{x_{e2}}), \tag{6}$$

$$\dot{n}_{ij} = \sum_{e=1}^{D_\alpha} \mathbb{1}(x_{e1} = i, x_{e2} = j, c_{e1} = c_{e2}), \qquad Y_{ij} = \mathbb{1}\left(\dot{n}_{ij} + \dot{n}_{ji} \geq 1\right). \tag{7}$$

Here $x_{ev} \in \{1, 2, \ldots\}$ for $v = 1, 2$ indicate the nodes associated with edge $e$ as in Eq. (5). An edge $Y_{ij}$ is then present in the undirected graph if and only if at least one multigraph edge is not thinned. Equivalently, Eq. (7) implies that the observed undirected graph is a binary projection of the multigraph edges that are retained after the thinning process.

To determine the distribution of $Y_{ij}$, note that the probability that nodes $i$ and $j$ are assigned to the same community (marginalizing across communities) equals $P(c_{ei} = c_{ej}) = \sum_{k=1}^{K} \pi_{ik}\pi_{jk}$. Therefore, marginalizing over the latent multigraph, $\dot{n}_{ij} \mid w_i, w_j, \pi_i, \pi_j \overset{\text{ind}}{\sim} \text{Poisson}(w_i w_j \sum_{k=1}^{K} \pi_{ik}\pi_{jk})$. The likelihood of the observed network thus equals

$$Y_{ij} \mid w_i, w_j \overset{\text{ind}}{\sim} \text{Bernoulli}\left(1 - \exp\left\{-2w_i w_j \sum_{k=1}^{K} \pi_{ik}\pi_{jk}\right\}\right). \tag{8}$$

Unlike the MMSB and GGP models, our *thinned generalized gamma process* (TGGP) model favors edges between nodes that have *both* large sociabilities and similar community memberships. The TGGP model is summarized graphically in Fig. 2 and illustrated in Figs. 1 and 3.

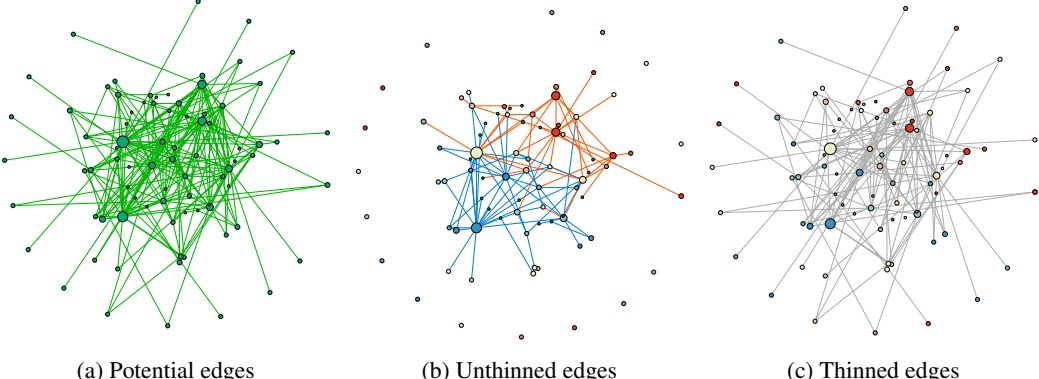

|  |  |  |
|---|---|---|
| (a) Potential edges | (b) Unthinned edges | (c) Thinned edges |

Figure 3: Visualization of networks simulated via a (thinned) GGP with $\alpha = 15, \sigma = 0.2, \tau = 1$. (a) Before thinning, *potential* edges are proposed according to nodes' sociabilities as in [25]. (b) After thinning, *unthinned* edges have colors (blue/red) corresponding to their assignment to $K = 2$ communities. (c) The *thinned* edges (gray) are those for which the connected nodes were assigned to different communities. In both (b) and (c), node colors represent their true community memberships, where lighter colors indicate more balanced memberships. Node sizes are proportional to betweenness centrality, and layout is determined by Gephi's Force Atlas 2 [42].

## 4   Monte Carlo Posterior Inference

In our TGGP model, the latent (unthinned) multigraph has a distribution (Eq. (5)) that depends only on the node sociability parameters $w_i$. The thinning process (Eqs. (6) and (7)) then depends only on the community membership vectors $\pi_i$. As can be deduced from the Markov properties of the directed graphical model in Fig. 2, given these latent multigraphs, the posterior distributions of $(w_1, w_2, \dots)$ and $(\pi_1, \pi_2, \dots)$ are conditionally independent. This design simplifies posterior inference, and in particular allows us to apply Theorem 6 of [25] to derive the posterior of nodes' sociabilities, provided we condition on the (latent) multigraph. We thus implement a Gibbs sampling strategy for posterior inference where we sample both thinned and retained edges in the latent multigraph, as well as their community assignments, to allow for efficient updates of other variables[1].

Some steps in the sampler are straightforwardly derived, considering that we can rely on the approach described in [25] to update node sociabilities and GGP hyperparameters (see the Appendix for a detailed outline). However, the implementation of the sampler requires some additional careful development – which we detail below – to sample the latent multigraph, due to the unobserved thinning process. Multigraph sampling is made more computationally efficient via model properties that are unique to sparse, as opposed to dense, stochastic block models. Letting (filled dot) $\dot{n}_{ij}$ denote the number of unthinned multigraph edges as in (7), and (empty dot) $\mathring{n}_{ij} = n_{ij} - \dot{n}_{ij}$ denote the number of thinned multigraph edges, we sample the latent multigraph as follows.

**Sampling of $\dot{\mathbf{n}}_{\mathbf{ij}}$:**   From Eq. (7), there exists an edge in the observed graph if and only if there is at least one unthinned edge between nodes $i$ and $j$ in the latent multigraph. Thus rather than considering each pair $i, j$, we only need to sample $\dot{n}_{ij}$ when $Y_{ij} = 1$, an operation with cost linear in the number of observed (undirected) edges. Conversely, $Y_{ij} = 0$ implies $\dot{n}_{ij} = \dot{n}_{ji} = 0$. Since there must be at least one unthinned edge when $Y_{ij} = 1$, the posterior follows a *zero-truncated* Poisson distribution:

$$(\dot{n}_{ij} + \dot{n}_{ji}) \mid Y_{ij} = 1, w_i, w_j, \pi_i, \pi_j \sim \text{Zero-Trunc-Poisson}\left(2 w_i w_k \sum_{k=1}^{K} \pi_{ik}\pi_{jk}\right).$$

For each unthinned edge we sample a single community assignment, since by definition these are the edges whose nodes have been assigned to the same communities. We can sample the community assignment of an unthinned edge between $i$ and $j$ easily as a draw from $\text{Cat}(\pi_{i1}\pi_{j1}, \dots, \pi_{iK}\pi_{jK})$.

**Sampling of $\mathring{\mathbf{n}}_{\mathbf{ij}}$:**   By construction, since thinned edges are unobserved, all pairs of nodes $(i, j)$ may have edges that are thinned. These edges are auxiliary variables necessary to obtain the full conditional posterior distribution of $W_\alpha$. Also, to update community memberships and global frequencies, it is necessary to assign every thinned edge to a pair of discordant (non-matching) communities. Thanks

---

[1] Code can be found on the first author's website https://federicazoe.github.io/

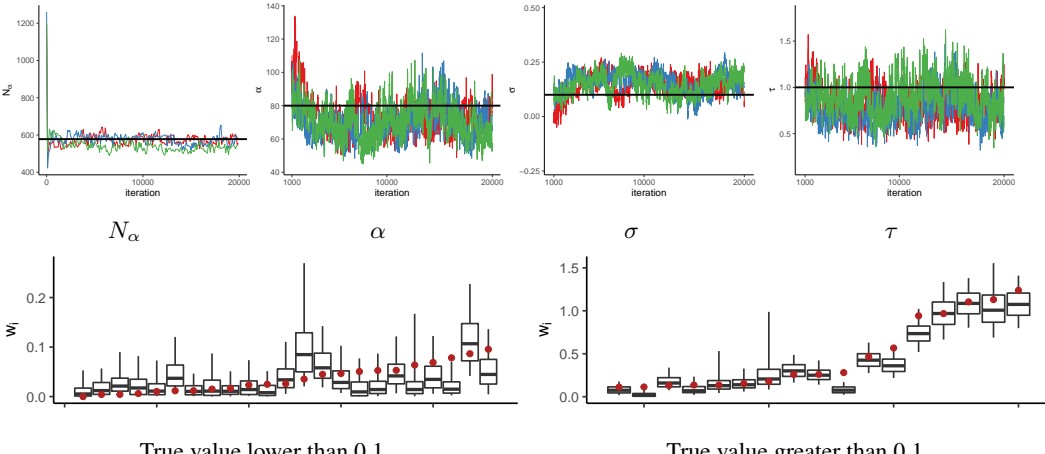

$$N_\alpha \qquad\qquad \alpha \qquad\qquad \sigma \qquad\qquad \tau$$

True value lower than 0.1                 True value greater than 0.1

Figure 4: Validation of the MCMC sampler on a graph with $K = 15$ communities, and $N = 386$ observed nodes, simulated from the TGGP. *Top*: Traceplots of $N_\alpha, \alpha, \sigma$, and $\tau$. Red, blue, and green indicate three distinct MCMC chains. Black horizontal lines mark the true values for the simulated graph. *Bottom*: True simulated values (red) and 95% credible intervals for the sociability parameters $w_i$ of 40 randomly selected nodes.

to the properties of Poisson processes, we do not need to sample $\mathring{n}_{ij}$ for all node pairs; we can develop a more efficient sampler that exploits the thinned model construction. We first sample a *proposed* total number of thinned edges $\mathring{D}_P \sim \text{Poisson}\left(2\sum_{i,j=1}^N w_i w_j\right)$. We then assign each of these independently to a pair of nodes according to their sociabilities as in Eq. (5), and subsequently to a pair of communities given the memberships of the assigned nodes as in Eq. (6). Finally, we determine $\mathring{n}_{ij}$ from the number of edges that were assigned to discordant communities.

**Sampling unobserved nodes:** As illustrated in Figs. 1 and 3, the total number of nodes $N_\alpha$ with at least one edge in the *unthinned* multigraph is likely to be different from the total number of nodes with at least one edge *after* thinning and – as a consequence – from the total number of nodes $N$ in the observed binary graph $Y$. There is thus some latent number $N_\alpha - N \geq 0$ of nodes whose edges with the observed nodes have all been thinned. In order to learn this number, our MCMC sampler includes an approximate update of $N_\alpha$ according to the mean ratio $N_\alpha/N$ from graphs simulated from the GGP prior given the latest samples of the hyperparameters $\alpha, \sigma, \tau$. We found that this approximate method leads to convergence of the empirical MCMC-based estimate of $N_\alpha$ across all simulated and real data that we considered (see Fig. 4). Sampling of the thinned edges from these unobserved nodes introduces some additional complications that we efficiently resolve as detailed in the Appendix.

**Sampling community memberships $\pi_i$ and global frequencies $\beta$:** From the conjugacy of Dirichlet priors to categorical likelihoods, node community memberships $\pi_i$ have closed-form Dirichlet posteriors $\pi_i \mid M_i, \beta \overset{\text{ind}}{\sim} \text{Dirichlet}(\zeta\beta_1 + M_{i1}, \ldots, \zeta\beta_K + M_{iK})$, where $M_{ik}$ equals the number of edges that node $i$ participated in while being assigned to community $k$. Given community assignments of nodes in the latent multigraph, the global community frequencies $\beta$ may be efficiently resampled using auxiliary-variable methods developed for the hierarchical Dirichlet process [6] (see Appendix). By learning $\beta$, node community memberships $\pi_i$ become sparse, placing significant probability mass only on a data-dependent subset of the full set of $K$ communities allocated in Eq. (1).

**Computational complexity:** For a $K$-community TGGP model of a graph with $E$ observed edges, resampling the latent multigraph requires $\mathcal{O}(EK)$ operations. (The thinned multigraph has fewer edges, and thus its resampling increases costs by a small constant.) Closed-form resampling of community memberships is faster, requiring only $\mathcal{O}(NK)$ operations for an $N$-node graph. The CGGP model [29] uses Hamiltonian Monte Carlo proposals with similar cost, but empirically the CGGP sampler mixes slower and has inferior performance (see experiments). By exploiting sparse matrices and parallelization, both samplers may be scaled to networks with tens of thousands of nodes; very-large networks may require alternative approximate inference algorithms.

**MCMC sampler validation:** We evaluate the proposed posterior inference method on a graph with 15 communities simulated from the TGGP model. The simulated graph had 1571 undirected, unthinned edges, $N = 386$ observed nodes (i.e., with at least one unthinned edge), and $N_\alpha - N = 192$

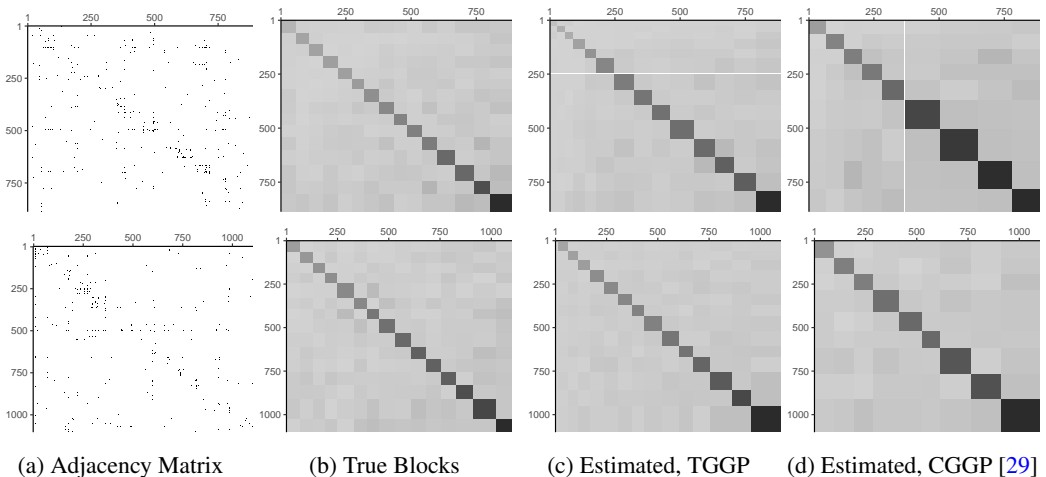

| (a) Adjacency Matrix | (b) True Blocks | (c) Estimated, TGGP | (d) Estimated, CGGP [29] |

Figure 5: Results on two simulation experiments: a graph generated with the CGGP [29] (top) and a graph simulated with the proposed TGGP (bottom). Column (a) shows the true adjacency matrices, sorting nodes into blocks according to the strongest community membership; (b) shows the relative edge density in the true blocks, sorting blocks by intensity. Columns (c) and (d) show the blocks estimated, respectively, with the TGGP and the CGGP. In both simulations, we generate data setting $\alpha = 250$, $\sigma = 0.1$, $\tau = 1$, and $K = 15$. For the CGGP we set $a_k = 1/K$ and $b_k = 1$; for the TGGP we set $\beta = (1/K, ..., 1/K)$ and $\zeta = 1$. Inference uses the true number of communities for the CGGP, while our TGGP is given a loose upper bound of $K = 50$, $\gamma = 10$, $\zeta = 0.2$. The TGGP closely approximates the true community structure, while the CGGP consistently underestimates the true number of communities.

unobserved nodes (i.e., with only thinned or self edges). Fig. 4 (top) shows traceplots from three separate MCMC chains, each of which was run for 20,000 iterations. The traceplot of the total number of nodes (observed + unobserved) $N_\alpha$ shows all values sampled since initialization to demonstrate that even very different initializations led the three MCMC chains to quickly concentrate around the same (true) value of $N_\alpha$. The traceplots of the GGP hyperparameters $\alpha$, $\sigma$, and $\tau$ show samples after convergence (which took roughly 1000 iterations) and demonstrate that the posterior distribution effectively concentrated around the true values of these hyperparameters. Fig. 4 (bottom) shows summaries of the posterior distributions of nodes' sociabilities for 40 randomly selected nodes spanning from low to relatively large sociability, suggesting that the TGGP model also effectively recovers nodes' sociability parameters.

## 5  Experimental Results

### 5.1  Simulation

We discuss a simulation study where we investigated the performance of our proposed TGGP model, and the CGGP model of [29], based on simulated data generated from either model.

First, we present the results from a sparse graph with 15 communities, simulated from our TGGP model by setting $\alpha = 250$, $\sigma = 0.1$, $\tau = 1$ for the distribution of node sociabilities and $\gamma = 10$, $\zeta = 0.2$ for the distribution of node community memberships. The adjacency matrix of the resulting undirected graph is shown in Figure 5(a, bottom). Figure 5(b, bottom) sorts the simulated nodes into blocks according to their main membership and plots the density of edges in each block, demonstrating that the simulated graph has a clear block structure. We then run our MCMC sampler for 50,000 iterations, discarding the first 40,000 samples as burn-in. For our model fitting, we let 50 be the upper bound to the number of communities and we set $\gamma = 10$ and $\zeta = 0.5$ to allow for learning the number of communities. From a qualitative comparison of the (b) and (c) bottom panes in Figure 5, we see that the community memberships recovered from our model are close to the underlying truth. In contrast, the CGGP model [29] struggles to recover the true community structure of the network (Fig. 5(d bottom)), despite being fitted with the number of communities $K = 15$ set to match the truth. We also simulated a graph from the CGGP model of [29] with parameters $a_k = 1/K$, $b_k = 1$ chosen to induce a community membership prior similar to our TGGP. We fit both the CGGP model

and the TGGP model to the generated data. When fitting with the CGGP we still set the number of communities equal to the truth ($K = 15$). The results plotted in Fig. 5(d) indicate that the block structures learned by our (misspecified) TGGP model are nevertheless closer to truth than those recovered by the CGGP. Additional results (shown in Appendix B) suggest that the parameters that represent the relative size of communities across the network are also more difficult to identify in the CGGP than in our TGGP formulation.

To quantify the superior community recovery of our TGGP model, recall that for each observed node the CGGP estimates a $K$-dim. positive real vector of community-specific sociabilities $(w_{i1}, \ldots, w_{iK})$, while our TGGP infers a $K$-dim. probability vector of community memberships $(\pi_{i1}, \ldots, \pi_{iK})$. For quantitative comparison, we standardize true and inferred sociability vectors by dividing each $w_{ik}$ by the sum $\sum_{h=1}^{K} w_{ih}$. Also, since the TGGP model was fitted with $K$ set to an upper-bound of $50$ rather than to the true number of communities, when analyzing TGGP results we pad the true vectors of community memberships (for TGGP-simulated data) and sociabilities (for CGGP-simulated data) with zeros. For estimated parameters, community indices are then permuted to maximize similarity with true parameters. Letting $\psi_i$ and $\hat{\psi}_i$ be, respectively, true and inferred community memberships/sociabilities, we de-

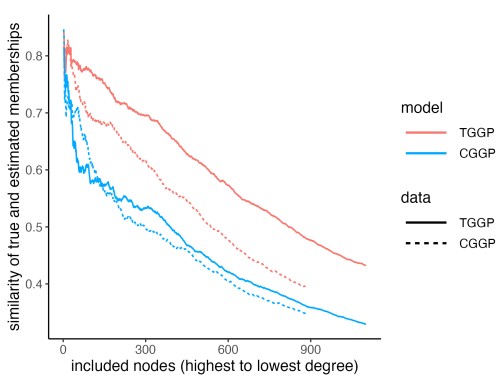

Figure 6: Plot of similarity between true and inferred community memberships for networks simulated (*data*) and fitted (*model*) with the TGGP and CGGP [29]. Our TGGP model uniformly achieves higher similarities.

fine similarity as $1 - \frac{1}{N} \sum_{i=1}^{N} \mathrm{d}(\psi_i, \hat{\psi}_i)$ where the distance is the L1 total variation $\mathrm{d}(\psi_i, \hat{\psi}_i) = \frac{1}{2} \sum_{k=1}^{K} |\psi_{ik} - \hat{\psi}_{ik}|$. Fig. 6 confirms that the TGGP more accurately recovers community memberships for both networks, and for nodes of both high and low degree.

## 5.2 Real network data

We compare the performance of our TGGP framework for adding overlapping communities to the GGP model with the approaches detailed in Sec. 2.2 and 2.1: the CGGP (sparse with mixed memberships) [29], the SBM-GGP model (sparse with single memberships) [28], and the more classical SBM (dense with single membership) [2] and MMSB (dense with mixed memberships) [5]. We run 50,000 iterations of the five models' MCMC on four real-world networks; see Appendix for data sources and pre-processing. Each model was fit to fully observed data to learn node-specific parameters (e.g., sociabilities and community memberships) and community-interaction probabilities, using the values from the last MCMC iteration. Two different measures of posterior predictive accuracy [43] were then used to assess the goodness of model fit.

For the first evaluation measure, we randomly selected 5% of the entries equal to 1 in the adjacency matrix and set them to 0. We then compute the probability of being 1 of all entries that are equal to 0 in the modified adjacency matrix to assess how well each model can distinguish the entries that should in fact be 1. This edge-retrieval measure is motivated by common applications of network analysis to recommendation tasks (suggesting which edges that are not observed should be present). Fig. 7 (top) plots recall (proportion of ones recovered) on the horizontal axis, and F-score (geometric mean of recall and precision, which is the proportion of ones correctly predicted among all entries predicted as ones) on the vertical axis. This retrieval task is very challenging because, for sparse networks, the missing edges are a tiny proportion of all zero entries in the modified adjacency matrix.

For the second evaluation measure, we randomly obscure 5% of *all* entries in the adjacency matrix to assess how well each model can recover whether an entry is equal to 0 or to 1. Fig. 7 (bottom) plots the receiver operating characteristic (ROC) curves, where false positive rate (number of zeros predicted as ones divided by the number of zeros) is on the horizontal axis, and true positive rate (proportion of ones correctly predicted as ones divided by the number of ones) is on the vertical axis. This second task has been used in many prior papers but is not as hard as the first one, because for a model to score well it suffices to learn small interaction probabilities, since almost all of the randomly

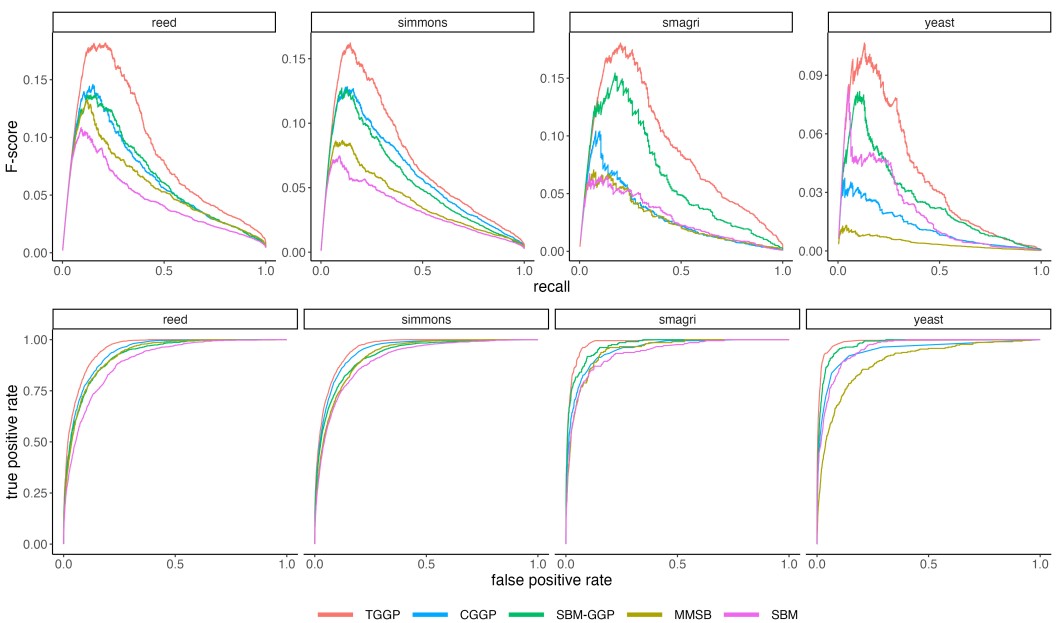

Figure 7: F-score versus recall for retrieval of missing edges (top), and ROC curves (bottom) for prediction of missing adjacency matrix entries, for four real-world networks of varying sparsity. Our TGGP uniformly outperforms the CGGP [29], SBM-GGP [28], MMSB [5], and SBM [2].

selected entries are 0. Fig. 7 shows that our TGGP model does consistently better than all baseline methods for all networks considered, according to both evaluation measures.

## 6 Discussion

We have proposed a framework for the analysis of binary network data that extends the GGP-based model of Caron and Fox [25] by allowing for overlapping community structures, depending both on the overall sociability of the nodes *and* the similarity of their community memberships. Our generative model uses a novel latent multi-graph framework where nodes can connect within the same or across different communities, but the edges formed across different communities are hidden (thinned) in the projection giving rise to the observed network. In contrast with alternative extensions of the original GGP network model [28, 29], our thinned generalized gamma process (TGGP) enables mixed memberships and facilitates regularization of the community distributions for each node, resulting in improved inference and reconstruction of the latent community structures. Also, our model allows for learning the number of communities directly from the data, by encouraging recovery of a set of non-empty communities smaller than a specified upper bound.

Monte Carlo inference for the TGGP scales linearly with the number of observed edges, and thus subquadratically with the number of nodes in sparse graphs. This leads to strong edge prediction performance for social and biological networks of moderate size. However, further inference innovations may be needed to scale to very-large networks with hundreds of thousands of nodes.

The proposed TGGP model is amenable to further extensions. As the adopted mixed membership framework relies on the well-studied Dirichlet-multinomial specification, the vast literature on hierarchical Dirichlet mixtures can be leveraged to capture more complex networks. It is possible to include additional node covariates or metadata to guide the allocation of nodes to communities. Finally, distributional and asymptotic properties of our model can be investigated by exploiting results related to finite mixtures and mixtures of finite mixtures [see, e.g., 34, 44, 45, 35].

## Acknowledgments and Disclosure of Funding

This research supported in part by NSF CAREER Award No. IIS1758028, and by the HPI Research Center in Machine Learning and Data Science at UC Irvine.

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
