# Appendix: Thinned random measures for sparse graphs with overlapping communities

## A    Monte Carlo Posterior Inference

We review the Gibbs sampling steps, some of which employ Metropolis-Hastings proposals rather than exact updates, for our thinned generalized gamma process (TGGP) model. We then describe each step in more detail.

**Gibbs sampling steps: Overview**

1. Update the edge counts and community assignments in the latent multigraph;
2. Update nodes' community memberships;
3. Update the global expected value of community memberships;
4. Update nodes' sociabilities;
5. Update the hyperparameters of the distribution of nodes' sociabilities;
6. Update the dimension of the latent multigraph.

### A.1    Updating edge counts and community assignments in the latent multigraph

We start by recalling that, in our TGGP model, the observed simple undirected graph is a binary projection of a latent (i.e. unobserved) directed multigraph. For each edge in which a node participates in the latent multigraph, the node is assigned to a community. Then, a given (non-self) edge is unthinned, and therefore present in the observed simple graph, if the communities that its two nodes are assigned to coincide; to recall this implication when describing the sampling strategy adopted for unthinned edges, we will refer to them as *within-community* edges. Otherwise, if the communities differ, the edge is thinned and therefore unobserved, and we will refer to such edges as *across-communities*. We note there may be self-edges in the latent multigraph. However, since we assume that self-edges are not observed irrespective of community assignments, no distinction is needed between within- and across- communities, and we refer to any of them simply as self-edges.

Two nodes have an edge in the observed undirected graph if there is at least one within-community edge between those nodes in the multigraph. *Observed nodes* are those that participate in at least one within-community edge. The nodes that in the multigraph are connected only within across-communities or self-edges are *unobserved nodes*. We call $N$ the number of observed nodes and we let $i = N + 1, \ldots, N_\alpha$ denote the indices of unobserved nodes.

To easily sample from the posterior of nodes' sociabilities we need (for both observed and unobserved nodes) each node's total degree, which equals the total number of edges in the multigraph (both within- and across- communities) in which the node participates. Also, to easily sample from the posterior of nodes' community memberships, we need their community assignment for each of the edges that they are connected to in the multigraph (both within- and across- communities). For both purposes, it suffices to track the entries of a single $N_\alpha \times K$ matrix of summary statistics $M$, where $M_{ik}$ equals the number of edges that node $i$ participated in while being assigned to community $k$. The total degree of node $i$ may then be obtained as $M_{i\bullet} = \sum_{k=1}^{K} M_{ik}$. We denote the counts of the assignments of a node $i$ to each community (that is, row $i$ of $M$) by $M_i = (M_{i1}, \ldots, M_{iK})$.

We will break down the sampling of the entries of $M$ in three steps: first edge counts and community assignments for within-community edges, second across-communities and self-edges between observed nodes, and finally across-communities and self-edges involving unobserved nodes. See Figure 1 for an illustration.

### A.1.1    Within-community edges

Recall that we assumed that an edge between two nodes is observed *if and only if* there is at least one within-community edge between those nodes in the latent multigraph. Then, the presence

of an edge between nodes $i$ and $j$ in the observed graph tells us that $\dot{n}_{ij} + \dot{n}_{ji} > 0$, that is the sum of within-community edges from $i$ to $j$ and from $j$ to $i$ in the latent directed multigraph is strictly positive. Similarly, the absence of an edge in the observed graph tells us that there is no within-community edge. Therefore, the number of within-community edges between two nodes only needs to be sampled for those pairs for which $Y_{ij} = 1$, $(i \geq j)$. An illustration is in Figure 1a. Because *a priori*, the number of within-community edges from $i$ to $j$ and from $j$ to $i$ have the same Poisson $\left( w_i w_j \sum_{k=1}^{K} \pi_{ik} \pi_{jk} \right)$ distribution, *a posteriori* we have

$$(\dot{n}_{ij} + \dot{n}_{ji}) \mid Y_{ij}, w_i, w_j, \pi_i, \pi_j \overset{\text{ind}}{\sim} \begin{cases} \text{0-Trunc-Poisson} \left( 2w_i w_j \sum_{k=1}^{K} \pi_{ik} \pi_{jk} \right) & \text{if } Y_{ij} = 1, \\ \delta_0 & \text{if } Y_{ij} = 0. \end{cases}$$

By definition, the community assignment of the nodes involved in a within-community edge coincide. So we need to sample a single community assignment valid for both nodes $i$ and $j$ for each of the $\dot{n}_{ij} + \dot{n}_{ji}$ within-community edges:

$$(z_{ij1}, \ldots, z_{ijK}) \mid \dot{n}_{ij} + \dot{n}_{ji}, \pi_i, \pi_j \sim \text{Multinomial} \left( \dot{n}_{ij} + \dot{n}_{ji}, \left( \frac{\pi_{i1} \pi_{j1}}{\sum_{k=1}^{K} \pi_{ik} \pi_{jk}}, \ldots, \frac{\pi_{iK} \pi_{jK}}{\sum_{k=1}^{K} \pi_{ik} \pi_{jk}} \right) \right),$$

and we increment $M_i = (M_{i1}, \ldots, M_{iK})$ and $M_j = (M_{j1}, \ldots, M_{jK})$ accordingly.

**Computational efficiency.** Note that the computational cost of sampling within-community edges is linear in the number of observed edges.

### A.1.2 Across-communities and self-edges between observed nodes

Because across-communities edges in the latent multigraph don't affect the observed simple graph, all pairs of observed nodes can have across-communities edges, both those pairs $i$ and $j$ for which $Y_{ij} = 1$ and those pairs $i'$ and $j'$ for which $Y_{i'j'} = 0$.

We thus need to sample the total number $\mathring{n}_{ij} + \mathring{n}_{ji}$ of across-communities edges for all pairs $i, j \in \{1, \ldots, N\}$ with $i \geq j$ (see Figure 1b). Because they are independent of the observed graph (conditional on nodes' sociabilities and memberships), the posterior distribution of across-communities edges is the same as their prior:

$$(\mathring{n}_{ij} + \mathring{n}_{ji}) \mid w_i, w_j, \pi_i, \pi_j \overset{\text{ind}}{\sim} \text{Poisson} \left( 2w_i w_j \left( 1 - \sum_{k=1}^{K} \pi_{ik} \pi_{jk} \right) \right).$$

However, rather than considering every pair of nodes, we can leverage the properties of independent Poisson-distributed random variables to draw $(\mathring{n}_{ij} + \mathring{n}_{ji})$ for all pairs of nodes from the correct distribution in a more efficient way. Specifically, we can first sample an auxiliary variable $\mathring{D}_P$ - representing a *proposed* number of across-communities edges between observed nodes - from the prior distribution on the total number of edges and self-edges involving only observed nodes:

$$\mathring{D}_P \mid w_1, \ldots, w_N \sim \text{Poisson} \left( 2 \sum_{i=2}^{N} \sum_{j<i} w_i w_j + \sum_{i=1}^{N} w_i^2 \right).$$

Then, we can proceed by associating each edge $e = 1, \ldots, \mathring{D}_P$ to a pair of nodes based only on nodes' sociabilities, and assigning each sampled node to a community based only on its own memberships:

$$x_{ev} \overset{\text{iid}}{\sim} \text{Cat} \left( \left( \frac{w_1}{\sum_{i=1}^{N} w_i}, \ldots, \frac{w_N}{\sum_{i=1}^{N} w_i} \right) \right), \qquad c_{ev} \mid x_{ev}, \pi_{x_{ev}} \overset{\text{ind}}{\sim} \text{Cat}(\pi_{x_{ev}}) \quad \text{for } v = 1, 2.$$

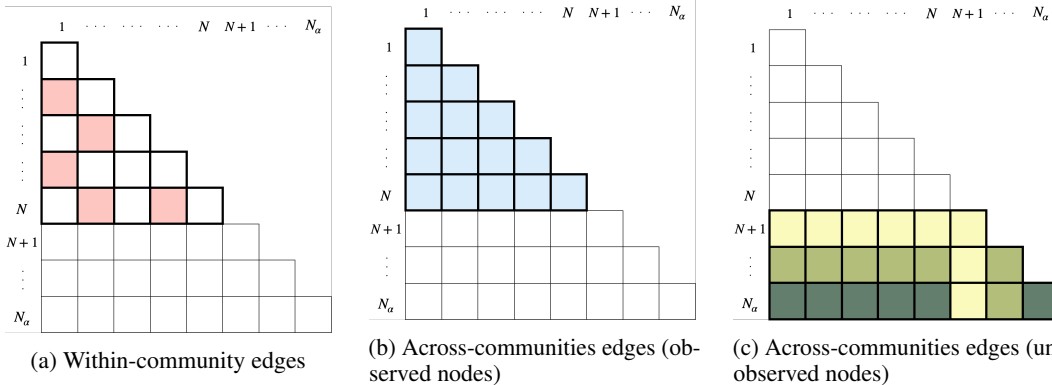

(a) Within-community edges     (b) Across-communities edges (observed nodes)     (c) Across-communities edges (unobserved nodes)

Figure 1: Illustration of the adjacency matrix resulting from an undirected version of the latent directed multigraph and corresponding sampling strategies. The entry located in row $i$ and column $j$ has the total (from $i$ to $j$, plus from $j$ to $i$) number for each of three types of edges. (a) *Within-community edges*: entries colored in pink mark pairs of nodes for which an edge is observed and for which within-community edges need to be sampled, as described in Sec. A.1.1. (b) *Across-communities and self-edges between observed nodes*: all entries corresponding to observed nodes are colored in blue to emphasize that these type of edges must be sampled for all those pairs. This can be done by proposing the total across these entries first, as described in Sec. A.1.2. (c) *Across-communities and self-edges involving unobserved nodes*: three different colors are used to mark which entries are sampled jointly according to the sequential sampling scheme described in Sec. A.1.3.

Finally, restricting to the resulting across-communities edges and self-edges, we have obtained a sample for them from the correct distribution. Accordingly, we increment the counts $(M_{i1}, \ldots, M_{iK})$ of community assignments for observed nodes $i = 1, \ldots, N$, i.e. the first $N$ rows of $M$.

**Computational efficiency.** Note that, using the strategy described in this subsection, the computational cost for sampling across-communities and self edges depends on the cost of assigning $\mathring{D}_P$ edges to a pair of nodes. Thanks to fast sampling techniques for sampling from discrete distributions like the alias method [46, 47, 48], this operation can be made to require only $\mathcal{O}(N \log N)$ computational time for pre-processing, after which edge assignments $x_{ev}$ may be generated in constant time.

### A.1.3    Across-communities and self edges involving unobserved nodes

Let $N_\alpha - N$ be the number of nodes that are active (have at least one connection) in the latent multigraph, but *not* in the observed simple graph. Let the labels of the nodes be ordered so that $i = N + 1, \ldots, N_\alpha$ correspond to the labels of these unobserved nodes. Conditional on these nodes not being observed but having connections in the latent multigraph, the information that we know is that each of them can only have across-communities or self-edges, but also that they *must* be associated to *at least one* across-communities or self-edge in the latent multigraph. This constraint implies that some care is required when sampling edges involving unobserved nodes. Specifically, we must generate the *edge counts and associated nodes*, that is simulate how many edges each unobserved node participates in, as well as what other nodes are associated to these edges. Given the nodes associated to an edge involving an unobserved node, community assignments can then be sampled quite easily, considering that they need to be discordant for non-self edges and that no restriction is applied to self-edges.

**Edge counts and associated nodes.** To correctly incorporate the constraint that each unobserved node must be associated to at least one across-communities or self-edge, the strategy that we adopt is to sample the total number of edges and the community assignments of unobserved nodes sequentially, starting from those involving node $N + 1$ with any other node (that is, the yellow entries in Figure 1c). Let $s_i$, for $i = N + 1, \ldots, N_\alpha$, be an auxiliary indicator variable that takes value 0 if node $i$ has been associated to an edge already and 1 otherwise. The variable $s_i$ thus indicates if the constraint that node $i$ must be associated to at least one edge is still active ($s_i = 1$) or is inactive ($s_i = 0$). As we start from node $(N + 1)$, all constraints are active and so $s_i = 1$ for all $i = N + 1, \ldots, N_\alpha$. We then sample the total number of edges $\mathring{n}_{(N+1)}$ between node $(N + 1)$ and any other node from

$$\mathring{n}_{(N+1)} \mid (w_1, \ldots, w_{N_\alpha}), (\pi_1, \ldots, \pi_{N_\alpha}) \sim \text{0-Trunc-Poisson} \left( 2w_{(N+1)} \mathring{p}_{(N+1)} + w_{(N+1)}^2 \right),$$

$$\mathring{p}_{(N+1)} = \sum_{j=1}^{N} \mathring{p}_{(N+1)j} + \sum_{j=N+2}^{N_\alpha} \mathring{p}_{(N+1)j}, \qquad \mathring{p}_{(N+1)j} = w_j \left( 1 - \sum_{k=1}^{K} \pi_{(N+1)k} \pi_{jk} \right).$$

Given the total number of edges associated with node $(N+1)$, letting $x_{e2}^{(i)}$ be the other node (possibly node $i$ itself) associated to edge $e = 1, \ldots, \mathring{n}_{(i)}$ of node $i$, we sample:

$$P(x_{e2}^{(N+1)} = j) \propto \begin{cases} \mathring{p}_{(N+1)j} & \text{if } j = 1, \ldots, N, N+2, \ldots, N_\alpha, \\ w_{(N+1)} & \text{if } j = N+1. \end{cases}$$

We then set $s_{(N+1)} = 0$ and also $s_i = 0$ for every node $i \in \{N+2, \ldots, N_\alpha\}$ that has been associated to some edges with node $(N+1)$.

Next, we can move on to sampling all the *remaining* edges involving node $N+2$, that is those with any other node *except* node $N+1$ ( the light green entries in Figure 1c). In general, when we sample edges for node $(N+d)$ for $d = 2, \ldots, (N_\alpha - N)$, there are two considerations we must keep into account: (1) the edges between node $(N+d)$ and nodes $(N+1), \ldots, (N+d-1)$ have already been sampled and so those nodes should not be re-considered; (2) if $s_{(N+d)} = 0$, the constraint on node $(N+d)$ is no longer active and we may possibly sample 0 additional edges. Putting these considerations together and letting $\mathring{n}_{(N+d)}$ be the number of remaining edges to sample when considering node $(N+d)$, that is those with nodes $\{1, \ldots, N, N+d, \ldots, N_\alpha\}$, we have

$$\mathring{n}_{(N+d)} \mid (w_1, \ldots, w_{N_\alpha}), (\pi_1, \ldots, \pi_{N_\alpha}), s_{(N+d)} \sim$$

$$\begin{cases} \text{0-Trunc-Poisson} \left( 2w_{(N+d)} \mathring{p}_{(N+d)} + w_{(N+d)}^2 \right) & \text{if } s_{(N+d)} = 1, \\ \text{Poisson} \left( 2w_{(N+d)} \mathring{p}_{(N+d)} + w_{(N+d)}^2 \right) & \text{if } s_{(N+d)} = 0, \end{cases}$$

where

$$\mathring{p}_{(N+d)} = \sum_{j=1}^{N} \mathring{p}_{(N+d)j} + \sum_{j=N+d+1}^{N_\alpha} \mathring{p}_{(N+d)j}, \qquad \mathring{p}_{(N+d)j} = w_j \left( 1 - \sum_{k=1}^{K} \pi_{(N+d)k} \pi_{jk} \right).$$

Then for $e = 1, \ldots, \mathring{n}_{(N+d)}$, we sample

$$P(x_{e2}^{(N+d)} = j) \propto \begin{cases} \mathring{p}_{(N+d)j}, & \text{if } j = 1, \ldots, N, N+d+1, \ldots, N_\alpha, \\ w_{(N+d)} & \text{if } j = N+d. \end{cases}$$

**Community assignments.** For every node $i = N+1, \ldots, N_\alpha$ with $\mathring{n}_i > 0$, and for each $e = 1, \ldots, \mathring{n}_i$, we sample a community assignment $c_{e1}^{(i)}$ for node $i$ given $x_{e2}^{(i)}$ from

$$P(c_{e1}^{(i)} = k \mid \pi_i, x_{e2}^{(i)}, \pi_{x_{e2}^{(i)}}) \propto \begin{cases} \pi_{ik} \left( 1 - \pi_{x_{e2}^{(i)}, k} \right), & x_{e2}^{(i)} \neq i, \\ \pi_{ik}, & x_{e2}^{(i)} = i, \end{cases}$$

for $k = 1, \ldots, K$. For non-self-edges, we rescale $\pi_{ik}$ to account for the knowledge that the community to which the other node associated to edge $e$ is assigned needs to differ from that to which node $i$ is assigned for this edge. We increment $M_{ic_{e1}^{(i)}}$ accordingly, and then given $c_{e1}^{(i)}$ and $x_{e2}^{(i)}$, we can sample a community assignment $c_{e2}^{(i)}$ for node $x_{e2}^{(i)}$ from

$$P(c_{e2}^{(i)} = \ell \mid c_{e1}^{(i)}, x_{e2}^{(i)}, \pi_{x_{e2}^{(i)}}) = \begin{cases} \dfrac{\pi_{x_{e2}^{(i)}\ell}}{(1 - \pi_{x_{e2}^{(i)}c_{e1}^{(i)}})}, & x_{e2}^{(i)} \neq i, \ \ell \in \{1, \dots K\} \backslash c_{e1}^{(i)}, \\ \pi_{i\ell}, & x_{e2}^{(i)} = i, \ \ell \in \{1, \dots K\}, \end{cases}$$

and then increment $M_{x_{e2}^{(i)}c_{e2}^{(i)}}$. Notice that for non-self-edges, we have removed the community sampled for node $i$ from the range of available communities for the assignment of node $x_{e2}^{(i)}$, and rescaled $\pi_{x_{e2}^{(i)}}$ accordingly.

**Computational efficiency.** In our TGGP model, unobserved nodes will have low sociability parameters; otherwise, they would be likely to be linked to unthinned edges. Thus $\mathring{n}_{(N+d)}$ tends to be small for all $d = 1, \dots, N_\alpha - N$. For sampling their edges and community assignments, it is just necessary to compute a $(N+1) \times N_\alpha$ matrix $\mathring{P}$ whose entries $\mathring{p}_{ij}$ represent the relative probability of associating node $j$ to an edge sampled for node $i$.

## A.2 Updating nodes' community memberships

Here we describe the update of $\pi_i = (\pi_{i1}, \dots, \pi_{iK})$, the distribution over community memberships of node $i = 1, \dots, N_\alpha$. Recall the prior on $\pi_i$:

$$\pi_i \mid \beta \overset{iid}{\sim} \text{Dirichlet}(\zeta\beta_1, \dots, \zeta\beta_K),$$

where $\beta = (\beta_1, \dots, \beta_K)$ is a vector of community frequencies that itself has a Dirichlet prior, and equals the expected value of $\pi_i$. For our experiments we set $\zeta = 0.5$, to encourage community memberships of individual nodes to be more polarized than $\beta$ and typically only use a subset of the available communities.

For each edge in the latent multigraph to which node $i$ is associated, the generative model has node $i$ being assigned to a community sampled independently from $(1, \dots, K)$ with probabilities $(\pi_{i1}, \dots, \pi_{iK})$, like in multinomial sampling. The Dirichlet and the multinomial are conjugate distributions, so the posterior of $\pi_i$ is easily obtained. Indeed, conditional on the concentration parameter $\beta$ and the $N_\alpha \times K$ matrix $M$ whose rows record the number of times $M_i = (M_{i1}, \dots M_{iK})$ that an edge $i = 1, \dots, N_\alpha$ was assigned to each community $k = 1, \dots, K$, the posterior distribution of the vectors of community memberships are obtained as:

$$\pi_i \mid M_i, \beta \overset{ind}{\sim} \text{Dirichlet}(\zeta\beta_1 + M_{i1}, \dots, \zeta\beta_K + M_{iK}).$$

An advantage of this specification is that it allows sparse community memberships, in that the sample $(\pi_{i1}, \dots, \pi_{iK})$ may have elements that are (almost) exactly equal to zero.

## A.3 Updating the global expected value of community memberships

The posterior distribution of the expected value $\beta = (\beta_1, \dots, \beta_K)$ of nodes' community memberships informs about summaries of the latent community structure of a network, such as the likely number of active communities and the relative size of different communities. Recall the prior on $\beta$:

$$\beta \sim \text{Dirichlet}\left(\frac{\gamma}{K}, \dots, \frac{\gamma}{K}\right),$$

where the value of $\gamma$ controls the sparsity of the prior on $\beta$, with values of $\gamma \ll K$ being more sparsity-inducing. In our experiments, we fix $\gamma = 10$ but $K = 50$. This hierarchical prior approaches the hierarchical Dirichlet process [6] in the limit as $K \to \infty$.

Here we illustrate a strategy for sampling from the posterior of $\beta = (\beta_1, \dots, \beta_K)$ that relies on auxiliary variables drawn according to the so-called "Chinese restaurant franchise" representation of a hierarchical Dirichlet model [6]. We can represent each node $i = 1, \dots, N_\alpha$ as a restaurant, where every new community assignment for node $i$ can be represented as a new customer entering

restaurant $i$ and ordering a dish (community) $k \in \{1, \ldots, K\}$. Letting $T_e^{(i)}$ be the number of tables already started at restaurant $i$ when a new customer $e = 1, 2, \ldots$ enters, customer $e$ decides to sit at an existing table $t \in \{1, \ldots, T_e^{(i)}\}$ with probability proportional to the number of customers currently sitting at that table (and eat the dish $k_t^{(i)}$ that was ordered by the first customer who sat at the table); or, they can start a new table $T_e^{(i)} + 1$ with probability proportional to $\zeta$, and pick a new dish $k_{T_e^{(i)}+1}^{(i)}$ from the menu according to $\beta$. Let $t_e^{(i)}$ be the table chosen when customer $e$ enters restaurant $i$, or equivalently the table chosen for the community assignment of node $i$ for when edge $e = 1, \ldots, M_{i\bullet}$ is associated to it. Then, letting $q_e^{(i)} = (q_{e1}^{(i)}, \ldots, q_{eT_e^{(i)}}^{(i)})$ be the number of customers already sitting at each of the tables when customer $e$ enters:

$$p(t_e^{(i)} = t \mid q_e^{(i)}) \propto \begin{cases} q_{et}^{(i)} & \text{if } t \in \{1, \ldots, T_e^{(i)}\}, \\ \zeta & \text{if } t = T_e^{(i)} + 1. \end{cases}$$

Conditional on $t_e^{(i)}$ and letting $\tilde{\kappa}_t^{(i)}$ be the dish served at table $t$ in restaurant $i$, the dish $\kappa_e^{(i)} \in \{1, \ldots, K\}$ ordered when customer $e$ enters restaurant $i$ is sampled according to

$$p(\kappa_e^{(i)} = k \mid t_e^{(i)} = t, \beta) = \begin{cases} 1 & \text{if } t \in \{1, \ldots, T_e^{(i)}\}, \kappa_e^{(i)} = \tilde{\kappa}_t^{(i)}, \\ 0 & \text{if } t \in \{1, \ldots, T_e^{(i)}\}, \kappa_e^{(i)} \neq \tilde{\kappa}_t^{(i)}, \\ \beta_k & \text{if } t = T_e^{(i)} + 1. \end{cases}$$

From the last sampling equation we see that the posterior distribution of $\beta$ depends on how many *tables* (rather than customers) are being served each dish across all restaurants in the franchise. Let $\bar{T}_k^{(i)}$ be the number of tables serving dish $k$ at restaurant $i$ and let $\bar{T}_{\bullet k} = \sum_{i=1}^{N_\alpha} \bar{T}_k^{(i)}$ be the total number of tables serving dish $k$ across all restaurants in the franchise. The summary statistics that we need for the posterior of $\beta$ is the vector $\bar{T} = (\bar{T}_{\bullet 1}, \ldots, \bar{T}_{\bullet K})$.

We then sample, for every restaurant $i = 1, \ldots, N_\alpha$ (i.e. for every node) and every dish $k = 1, \ldots, K$ (i.e. for every community), the number of tables $\bar{T}_k^{(i)} \leq M_{ik}$ occupied by the $M_{ik}$ customers eating dish $k$ at restaurant $i$. Note that $M_{ik}$ is the entry in row $i$ and column $k$ of the $N_\alpha \times K$ matrix $M$ sampled at step 1 (see Section A.1).

Because the distribution of table arrangements is invariant to the order in which customers enter, we can assign one at a time each of the $e = 1, \ldots, M_{ik}$ customers eating dish $k$ at restaurant $i$ to either sit at an existing table ($\tilde{t}_{ek}^{(i)} = 0$) or to start a new table ($\tilde{t}_{ek}^{(i)} = 1$) according to:

$$\tilde{t}_{ek}^{(i)} \mid \beta \sim \text{Bernoulli}\left(\frac{\zeta \beta_k}{(e-1) + \zeta \beta_k}\right),$$

where we have $\zeta \beta_k$ since we are conditioning on the customer being served dish $k$ and we weight $\zeta \beta_k$ with respect to $(e-1)$ because, when considering customer $e$, there are already $(e-1)$ customers sitting at some table serving dish $k$. Thus, the chances that customer $e$ sits at one of the existing tables is proportional to $(e-1)$. Finally, we set $\bar{T}_k^{(i)} = \sum_{e=1}^{M_{ik}} \tilde{t}_{ek}^{(i)}$.

After sampling the auxiliary variables, we can easily update $\beta$ according to

$$\beta \mid \bar{T} \sim \text{Dirichlet}\left(\frac{\gamma}{K} + \bar{T}_{\bullet 1}, \ldots, \frac{\gamma}{K} + \bar{T}_{\bullet K}\right).$$

### A.4  Updating nodes' sociabilities

Let $M_{i\bullet} = \sum_{k=1}^{K} M_{ik}$ be the total degree of node $i$ (whose sampling was described in Section A.1). Conditional on the vector of total degrees $(M_{1\bullet}, \ldots, M_{N_\alpha \bullet})$ and on the hyperparameters $\alpha, \sigma$ and $\tau$, the update of nodes' sociabilities $(w_1, \ldots, w_{N_\alpha})$ can be done via a Hamiltonian Monte Carlo step [49, 50] as detailed by Caron and Fox in Section 7.2 and in Appendix F.1 of [25].

### A.5 Updating hyperparameters of sociabilities' distribution

We assume the same improper priors on the hyperparameters $\alpha, \sigma$, and $\tau$ as in [25]:

$$P(\alpha) \propto \frac{1}{\alpha},$$
$$P(\sigma) \propto \frac{1}{1-\sigma},$$
$$P(\tau) \propto \frac{1}{\tau}.$$

To update these hyperparameters we follow the approach that Caron and Fox describe in Section 7.2 and in Appendix F.2 in [25].

### A.6 Updating the dimension of the latent multigraph

Even though our interest lies in the sociabilities and community memberships of observed nodes $i = 1, \ldots, N$, to derive their exact posterior we need to also consider the unobserved nodes $N+1, \ldots, N_\alpha$, that is those nodes whose edges in the latent multigraph are all thinned, being all either across-communities or self-edges. The distribution of the number of nodes $N_\alpha$ with at least one edge in the latent multigraph is not tractable, and we thus resort to an approximate method for updating $N_\alpha$.

Our update of $N_\alpha$ is based on sampling $Q$ multigraphs from the GGP prior evaluated at the current values of $\alpha, \sigma$ and $\tau$. For simulated graph $q$, let $N^{(q)}$ be the number of nodes associated to at least one within-community edge, and $N_\alpha^{(q)}$ be the number of nodes associated to at least one edge (of any type). We then compute the average ratio $r_Q = \frac{1}{Q} \sum_{q=1}^{Q} \frac{N_\alpha^{(q)}}{N^{(q)}}$ and set the new value of $N_\alpha^{(\text{new})} = N r_Q$. If the new value $N_\alpha^{(\text{new})}$ is smaller than the older value $N_\alpha^{(\text{old})}$, we keep the $N_\alpha^{(\text{new})}$ unobserved nodes that currently have the largest sociabilities, along with their current community memberships. If $N_\alpha^{(\text{new})} > N_\alpha^{(\text{old})}$, we keep all of the current $N_\alpha^{(\text{old})}$ unobserved nodes, sample the sociabilities for the remaining $N_\alpha^{(\text{new})} - N_\alpha^{(\text{old})}$ nodes by resampling from the sociabilities of the current nodes, and assign community memberships from the prior evaluated at the current value of $\beta$. Our experiments on simulated data suggest that, by updating $N_\alpha$ every 100 iterations and setting $Q = 10$, this method allows MCMC chains to concentrate around the true value of $N_\alpha$ and of the hyperparameters $\alpha, \sigma$, and $\tau$ (see Figure 4 (top left) in the manuscript).

## B  Additional results on simulated data

In Section 3 of the manuscript we observed that our TGGP specification is more regularized, and thus may be more easily identified, than the compound GGP formulation of [29]. Recall that, in our specification, the community memberships for node $i$ are a probability vector $\pi_i$ over $K$ communities generated from

$$\pi_i \mid \beta \overset{\text{iid}}{\sim} \text{Dirichlet}(\zeta\beta_1, \ldots, \zeta\beta_K), \quad i = 1, \ldots, N_\alpha.$$

In contrast, for the compound GGP the membership of node $i$ in community $k$ is modeled as

$$w_{ik} = w_{i0}\psi_{ik}, \quad \psi_{ik} \overset{\text{ind}}{\sim} \text{Gamma}(a_k, b_k), \qquad \text{for } i = 1, \ldots, N_\alpha, \ k = 1, \ldots, K,$$

where nodes' baseline sociabilities $\{w_{i0}\}_{i=1}^{N_\alpha}$ are drawn according to a $\alpha$-truncated generalized gamma process [37, 38, 39] with parameters $\tau \in (0, \infty)$, and $\sigma \in (-\infty, 1)$. In Section 5.1 of the manuscript, we noted how our results suggested that the TGGP model may be better at recovering the underlying community structure of both TGGP and CGGP simulated graphs. Here we provide additional insights into this comparison. Figures 2a and 2b show that, for a CGGP-simulated graph with 15 communities, $a_k = 1/15$ and $b_k = 1$ for all $k = 1, \ldots, 15$, the samples $(a_1, \ldots, a_{15})$ from the posterior estimated fitting the CGGP model are not concentrated around the true value or $1/15$,

and several among $\tau b_1, \ldots, \tau b_K$ (which according to [29] is better identifiable than $b_1, \ldots, b_K$) are considerably larger than their true value of 1. Conversely, Figure 2c shows that the TGGP model fitted with $K = 50$ correctly concentrates $\beta_k$ around 0 for most $k = 1, \ldots, K$ and, for the remaining communities, it tends to draw samples around the true value of $1/15$ (the simulated graph for the TGGP has 15 communities and a true value $\beta_k = 1/15$ for each of them).

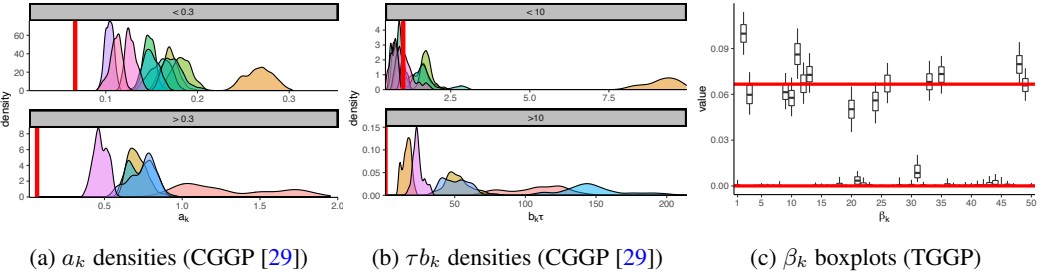

(a) $a_k$ densities (CGGP [29])     (b) $\tau b_k$ densities (CGGP [29])     (c) $\beta_k$ boxplots (TGGP)

Figure 2: Summaries of the posteriors of the parameters controlling the distribution of community memberships. (a) densities of $a_1, \ldots, a_K$ and (b) density of $\tau b_1, \ldots, \tau b_K$ (which according to [29] is better identifiable than $b_1, \ldots, b_K$) for the CGGP-simulated graph fitted with the CGGP model. Different communities correspond to different colors, red lines mark the true values, and the range of $a_k$ and $b_k$ are split between relatively small and large values to facilitate the plotting. (c) boxplots of the elements of the expected value $\beta$ of nodes' community memberships for the TGGP-simulated graph fitted with the TGGP model; a red line marks the value $1/15$ that is the true value for 15 among the 50 fitted $(\beta_1, \ldots, \beta_5 0)$, and another red line marks the value 0 that corrresponds to the true value for the remaining ones. Both the CGGP and TGGP models were fitted by running 50,000 iterations of the respective samplers and discarding the first 40,000 iterations as burn-in. Both the CGGP and TGGP simulated graphs had 15 underlying communities and hyperparameter values set to $\alpha = 250, \sigma = 0.1, \tau = 1$. The CGGP model was fitted setting the number of communities $K = 15$ equal to the truth, while the TGGP model was fitted with K = 50 and a sparsity-inducing prior on $\beta$.

## C    Real-data networks: preparation and references

The four real-data networks used in our experiments were pre-processed by extracting the main component for the networks, by removing self-edges and by making the observed network undirected. Summary information for each of these networks is displayed in Table 1.

| Network name | Type | Reference | Number of nodes | Number of edges |
|---|---|---|---|---|
| Reed | Online social network | [51] | 962 | 18812 |
| Simmons | Online social network | [51] | 1510 | 32984 |
| SmaGri | Co-authorship network | [52] | 1024 | 4916 |
| Yeast | Protein interaction network | [53] | 2224 | 6609 |

Table 1: Information on datasets used for real-data experiments

The networks Reed and Simmons were downloaded from `https://archive.org/details/oxford-2005-facebook-matrix`, the network SmaGri was downloaded from `https://www.cise.ufl.edu/research/sparse/matrices/Pajek/SmaGri.html`, and the network Yeast was obtained from `http://vlado.fmf.uni-lj.si/pub/networks/data/bio/Yeast/Yeast.htm`.