# OpenReview forum: "Thinned random measures for sparse graphs with overlapping communities"
_NeurIPS.cc/2022/Conference — NeurIPS 2022 Accept_

### Official Review · Reviewer_ju2a · 2022-07-10

**Rating:** 6
**Confidence:** 2
**Soundness:** 3 good
**Presentation:** 2 fair
**Contribution:** 2 fair

**Summary:**

This paper proposed a new framework for analyzing the community structure in the network. They modeled the overlapping community structures in the network by extending the generalized gamma process(GGP) model.
The model first generated a latent multigraph where edges are assigned depending on the sociability of nodes and their community memberships. Then, the edges in the multigraph that connect two different communities are thinned (unobservable). The observed network is formulated as the projection of the latent multigraph excluding the thinned edges.
They also provided an efficient way to implement the posterior inference by utilizing the sparse property of the observed network. Specifically, they sampled the community concordant and community discordant edges separately based on the observed network.
Their experiments also showed that this new model recovered the community membership in the network better than previous model.

**Questions:**

1. I am a little confused about the conclusion that "model formulation allows for learning the number of communities directly from the data, as it does not require fixing this number a priori. " In the posterior inference, you still need the number of communities K in the computation. Do you mean that this number K can be chosen and changed, not fixed?

2. In section 2.2, you mentioned that for fixed parameters, SBM and MMSB may only generate dense graphs. While many previous works on SBM and MMSB considered those parameters depending on the number of nodes like O(1/n) or O(log n/ n), which can also provide sparse networks. I am just curious about how this work compares to those works with non-constant parameters.



**Limitations:**

Yes, they mentioned some potential generalizations of their model by including additional available information.

**Strengths And Weaknesses:**

This new framework proposed in this paper is interesting. This new model can infer the overlapping community structures in the network, which has many applications and attracts a lot of attention in the research. This new model captures the sparsity and mixed community membership in real-world networks.
This new model extended the generalized gamma process(GGP) model by introducing the mixed community membership for each node. The technique used in this extension is in some sense standard by utilizing the sparsity in the observed network.

---

> ### Author Response · Authors · 2022-08-02
> **Response to reviewer ju2a**
>
> Questions
>
> 1) The number K in our model represents an upper bound to the number of communities. Given this upper bound, the hierarchical dirichlet formulation that we employ, which is connected to the dirichlet process  [18], encourages the posterior distribution to concentrate on a smaller set S* of communities with cardinality less than K by favoring sparse vectors of community frequencies if a smaller number of communities is sufficient to model the data well. We can define this set as S* = {k* in {1,...,K} such that the overall frequency of the respective communities beta_k* is greater than some threshold (e.g. 0.01). In this sense, the number of communities can be learned from the data. The formulation of Todeschini does not have similar properties that ensure that it will learn a number of communities smaller than the specified K when this is supported by the data. We have made remarks in the manuscript (lines 65 - 70; lines 210 - 212).
>
> 2) By making the SBM and MMSB parameters depend on the number of nodes it is possible to make the model compliant with sparsity. But there are some drawbacks. Most importantly, a desirable statistical property called projectivity is lost: the marginal distribution of a subset of nodes with cardinality (N’ = N - R) can no longer be obtained marginalizing away the R removed nodes from the distribution of all N nodes. Another drawback is that the level of sparsity needs to be tuned by the researcher and cannot be learned from the data as with the GGP-based models. Finally a drawback of the dense SBM and MMSB models is that their MCMC samplers need to sample community assignments for every pair of nodes (also those with no edge), regardless of whether there is an adjustment of their parameters to achieve sparsity, thus the computation scales quadratically with the number of nodes.

---

> > ### Comment · Reviewer_ju2a · 2022-08-08
> > **To authors**
> >
> > Thanks for the response. My questions are well addressed.

---

### Official Review · Reviewer_3Qbu · 2022-07-11

**Rating:** 4
**Confidence:** 2
**Soundness:** 2 fair
**Presentation:** 1 poor
**Contribution:** 2 fair

**Summary:**

The paper proposes TGGP, a new model for sparse random graphs with planted overlapping communities, based on the Generalized Gamma Process (GGP), and shows some experiments comparing the behaviour of TGGP compared to the previously proposed CGGP. Compared to CGGP, the proposed TGGP does not require to fix the number of communities a priori and has superior performance in terms of community detection and prediction of missing edges on synthetic and real-world graphs.

**Questions:**

There are several parts in the paper that require an improved presentation. Some specific examples include:

L1: What are "exchangeable arrays"? This concept is not used elsewhere in the paper

L39: An entire paragraph is missing.

L42: $Y. Y_{ij}$; $.$ should probably be a $:$

The probability distributions Cat, Beta, etc. should be introduced briefly

L70-74: It is not clear how the $W_\alpha$ are generated. How does $\ell_i$ depend on $w_i$? Also, what exactly is $W_\alpha$?

Figure 1: Poisson rate should be $w_iw_j$ not $2w_iw_j$ according to Eq(4). Color bars are needed to understand the range of values of the colored cells

Figure 2: Little to no explanation for this figure is provided in the text

L112: $w_iw_j$ should not have bold font

L114: what are $a_k,b_k$?

L117: what is a non-matrix?

Figure 4: If CGGP requires to specify the communities, why do we then get 9 communities instead of 15? Also, the qualitative comparison should be accompanied by some quantitative metric

Figure 6: F-score, recall, ROC curve should be described better in the text: what exactly are you plotting here?

L182: what exactly is "your MCMC"?





**Limitations:**

The paper does not discuss potential limitation

**Strengths And Weaknesses:**

S.
* The idea of allowing for overlapping communities via thinning in the GGP model is interesting and new.
* TGGP performs better as compared to CGGP

W.
*  The paper seems more like a draft, lots of work is needed to improve the presentation. For example, more explanation of backgrounds and related work, better structured contents, and more clear presentation of the generative algorithm based on MCMC
* The performance in terms of community extraction / graph reconstruction is superior with respect to CGGP but it does not seem competitive with other available techniques for the specific tasks. So the authors should clarify what the main contribution exactly are, in particular to what extent they fit the machine learning community.
* The work should come with an accompanying code that can be used to reproduce the experiments shown in the manuscript and to allow the use of the random graph generator by future researchers

---

> ### Author Response · Authors · 2022-08-02
> **Response to reviewer 3Qbu**
>
> Weaknesses
>
> 1) We thank the reviewer for pointing out some ambiguities in the presentation. We made fixes based on the errors listed by the reviewer along with other changes to improve the overall presentation, as can be seen in the revised manuscript.
>
> 2) The stochastic block model (SBM) [2] is regarded as “*the most widely adopted model of networks with communities*” (Menczer, F., Fortunato, S. and Davis, C.A., A first course in network science. Cambridge University Press, 2020.). In the Appendix we compared our model (TGGP) not only to its closest alternative (CGGP), but also to a dense model with single community membership (SSB), a dense SBM with mixed-membership (MMSB), and a sparse single-membership (SBM-GGP) models. Our TGGP uniformly outperforms all of these baselines. To better highlight the contributions of our model, Figure 6 in the revised version of our manuscript shows a comparison with all aforementioned models rather than only with the CGGP.
>
> 3) We agree that accompanying code is necessary to allow other researchers to use our model and we intend to make the code publicly available if the manuscript is accepted.
>
> Questions
>
> 1) Networks of edges between N nodes can be represented in terms of their N x N adjacency matrix. Assuming that the adjacency matrix is exchangeable means that the distribution of its random elements is invariant to permutations of the set of node indices {1,...,N}. In practice, this implies that statistical inference does not depend on nodes’ indices, which is desirable as these indices are usually arbitrary. We thank the reviewer for pointing out that this notion was only mentioned in the abstract and we have added a reference to it in the introduction (lines 29 - 31).
>
> 2) The missing paragraph in the introduction was completed (lines 41 - 48).
>
> 3) The sentences were correct but the proximity of mathematical symbols may have created confusion. We rearranged the second sentence to improve readability (line 51).
>
> 4) Where “Cat” is first mentioned in the paper (line 55) we have clarified its meaning. The probability distributions Beta/Bernoulli/etc. are standard in this literature, and given NeurIPS page constraints, a detailed introduction is not feasible.
>
> 5) W_alpha is the set of potential nodes’ sociabilities w_i. These are generated using the jumps of a completely random measure and endowing the jumps (sociabilities) with a (independently sampled) random location on the real line. Among all jumps, only those with location smaller than a (hyperparameter) alpha are retained as potential nodes, forming the set W_alpha. We modified the manuscript to improve clarity (lines 82 - 88).
>
> 6) Your observation about the rate in the legend is right, thank you for pointing it out. We included color bars in the figure to clarify what colors represent and what is “low” or “high” (we prefer not to include numeric values in the legend as this is only a toy example for illustrating the model at a high-level).
>
> 7) We extended the explanation of Figure 2 and of its purpose in the text (lines 108 - 110 and lines 169 - 171).
>
> 8) The font of w_i and w_j was fixed.
>
> 9) a_k, b_k are the shape and rate of the Gamma distribution of the random variable beta_k in the CGGP model. We made this explicit (lines 132 - 134) and changed notation from beta_ik to psi_ik to avoid confusion between beta_ik of the CGGP model and beta_k of our TGGP model.
>
> 10) We meant to write “non-negative matrix” rather than “non-matrix”, thanks for finding this typo.
>
> 11) In that plot, blocks are defined according to the community where nodes have the greatest inferred community sociability w_ik. For the CGGP there were only 9 blocks because for none of the nodes the k* such w_ik* > w_ih for every h≠k* corresponded to one of the other 6 communities. We have now also included a quantitative metric that corroborates the qualitative results, thank you for the suggestion! (lines 255 - 266; Figure 5)
>
> 12) Figure 6 (top) plots recall (proportion of ones recovered) on the horizontal axis and F-score (geometric average of recall and precision, which is the proportion of ones correctly predicted as ones among all entries predicted as ones) on the vertical axis. Figure 6 (bottom) plots the ROC curves, that is false positive rate (number of zeros predicted as ones divided by the number of zeros) on the horizontal axis and true positive rate (proportion of ones correctly predicted as ones divided by the number of ones) on the vertical axis. We included this explanation in the manuscript (lines 277 - 293).
>
> 13) By “our MCMC” we meant the MCMC sampler that we use for inference; we made this clearer (line 201).

---

> > ### Comment · Reviewer_3Qbu · 2022-08-08
> > **Thanks**
> >
> > Dear author(s), thanks for your responses. I am quite happy with your changes, several of the issues I have pointed out are now fixed. I just want to point out that the code should be made available to the reviewers as well so that we can have the chance to verify the experimental claims by running the code ourselves.

---

### Official Review · Reviewer_oHB1 · 2022-07-12

**Rating:** 6
**Confidence:** 4
**Soundness:** 3 good
**Presentation:** 3 good
**Contribution:** 3 good

**Summary:**

The authors propose the thinned generalized gamma process (TGGP) for generating sparse graphs with mixed-membership block structure. It builds upon the GGP-based model of Caron and Fox \[12\] and recent extensions to incorporate block structure. The TGGP thins potential edges from a GGP by removing edges where the community assignments for two nodes do not match. The authors claim that their TGGP is superior to the compound GGP approach of Todeschini et al. \[17\], and empirical results support this claim.

**Questions:**

1. I don't understand the first task in the real network data section: "what entries of the adjacency matrix incorrectly classified as 0 are most likely to be edges". What does this mean? Are you randomly setting some of the 1's in the adjacency matrix to 0 first and then trying to find these entries?
2. The authors claim that the proposed inference procedure is computationally efficient. How does it compare to the CGGP? How large of networks can these models realistically scale up to? Is 10,000 nodes and 100,000 edges feasible?

**Limitations:**

Limitations are not really discussed, just possible extensions for further work. The authors should discuss the scalability by providing a rough scale of how large of networks they can fit in a reasonable amount of computation time (e.g. a few days).

**Strengths And Weaknesses:**

Strengths:
- Empirical results suggest that proposed TGGP provides a superior approach for modeling sparsity with mixed-membership block structure compared to the CGGP, which is the only other approach that can simultaneously model both properties.
- The paper is mostly well written, and the explanations are quite clear. I enjoyed reading it! However, it seems to have been written in a rush, with lots of sloppy presentation issues (see weaknesses) that detract from overall presentation quality.
- Proposed model and posterior inference algorithm appear to be technically sound and require some clever ways of sampling to avoid blowing up the computational complexity.

Weaknesses:
- Lots of presentation issues--this paper badly needs a copy edit and just a bit more time to correct the errors. I have provided a list below. These errors detract from what was a well written paper.
- Experiments are on somewhat small data sets, with a maximum of about 2,000 nodes and about 33,000 edges. Also, no experiments were performed on real data with labeled communities, so estimated communities are only compared based on simulated data.
- Novelty is only moderate, as the proposed model addresses the same setting as the CGGP from Todeschini et al. \[17\], and most of the results follow from the GPP-based construction of Caron and Fox \[12\].

Presentation issues:
- Unfinished last paragraph of intro beginning on line 39: "In this paper, we propose a novel random graph model that"
- The variable $x$ is first used in equation (5) in Section 2.2 but not defined until its later usage in Section 3.
- Lines 150-151: total number of proposed nodes $n_{ij}$ between any two edges $i$ and $j$ in the multigraph -> total number of proposed *edges* $n_{ij}$ between any two *nodes* $i$ and $j$ in the multigraph
- Line 184: estiamte -> estimate
- Figure 5 is too short--I cannot see how frequently the true simulated sociability values fall within the 95% credible intervals for most of the nodes, which have $w_i < 0.5$.
- Figure 6 should have axis labels rather than needing to read the caption. Also, I find Figure 4 in the supplementary material, which shows the same results as Figure 6 in the main paper, but with all 5 models being compared, to be more convincing. I suggest for the authors to replace Figure 6 in the main paper with Figure 4 in the supplementary.
- Checklist was not completed: entries 4 and 5 still listed as TODO.

---

> ### Author Response · Authors · 2022-08-02
> **Response to reviewer oHB1**
>
> Weaknesses
>
> 1) We thank the reviewer for pointing out some ambiguities in the presentation. We made fixes based on the errors listed by the reviewer along with other changes to improve the overall presentation, as can be seen in the revised manuscript.
>
> 2) We note that the networks that we used in our experiments are larger than those used in [17], and of comparable size as those in [16]; we believe our results already convincingly demonstrate the advantages of our proposed TGGP model.
>
> 3) We agree that it would be interesting to apply the model to real data with labeled communities. However, public network datasets of even moderate size very rarely include ground-truth communities, especially mixed-membership communities. In Section 5.1 we had included an experiment on simulated toy data where labels are known and we have now enriched it by computing similarity scores between true and estimated community memberships (see lines 255 - 266). We also note that most of the literature on network models with communities (including [7, 16, 17]) analyzes real-world datasets with no ground-truth communities, and uses qualitative analysis and/or predictive performance (as in our Sec. 5.2) to assess the quality of estimated communities.
>
> 4) While it is true that our model extends the GGP-based construction of Caron and Fox [12], differently from Todeschini [17] our model allows to learn the number of communities and paves the way for extensions based on the extensive available literature on the Dirichlet process, such as the inclusion of covariates in the model. Note that even without these extensions, performance gains over [17] are substantial. Also, the construction of the model via thinning of a random measure is rather new and could inspire future work even beyond the literature on networks.
>
> Presentation issues
>
> 1) We improved former Figure 5 by making two separate plots for nodes with w_i < 0.1 and w_i > 0.1. Thanks for pointing out a shortcoming of this figure. Note that to make space for a new plot of similarity scores between true and estimated memberships, we moved former Figure 5 to the appendix, Figure 4.
>
> 2) We replaced Figure 6 with (former) Figure 4 from the appendix, and we added axis labels.
>
> Questions
>
> 1) Yes, your understanding is correct. We randomly select 5% of the entries equal to 1 and set them equal to zero. Then we use the estimated node-specific parameters (i.e. sociabilities and community memberships) and community-interaction probabilities to compute the probability to be 1 of all entries that are equal to 0. We have explained both posterior predictive tasks more clearly in the revised manuscript (lines 277 - 293).
>
> 2) Let E, N and K be, respectively, the number of edges, nodes and communities. For the update of the latent multigraph, the computational cost for the CGGP is of the order of O(E x K), for the TGGP is less than 2 times O(E x K) (since the TGGP needs to update the thinned multigraph, but the cost of this can be contained by adopting the strategy that we proposed in Section 4). For the update of community memberships and sociabilities, both methods require computations of the order O(N x K), but the TGGP samples from the full conditional posterior while the CGGP uses a Hamiltonian Monte Carlo proposal, which tends to have slower convergence rates due to possible rejection of proposed updates and of correlated samples. Both models, exploiting sparse matrices and using parallelization where possible, can realistically scale up to (i.e. fit in up to one/two days) networks with tens of thousands of nodes. For larger networks, MCMC would likely need to be replaced by approximate posterior inference strategies, the development of which can be the object of future research. We have added to Section 4 a paragraph dedicated to computational cost (lines 213 - 220).

---

> > ### Comment · Reviewer_oHB1 · 2022-08-09
> > **Thank you**
> >
> > I thank the authors for providing the detailed responses to reviewers. I still support this paper for acceptance. The main weaknesses I see are that novelty and evaluation are both somewhat limited.

---

### Meta-Review · Area_Chair_Aa4w · 2022-08-27

**Recommendation:** Accept
**Confidence:** Less certain

**Metareview:**

The authors present a framework for thinning edges from from random graph realizations from the generalized gamma process (GGP) to generate sparse graphs with mixed community memberships. The authors provide an efficient Monte Carlo methods that scale sub-quadratically with the number of nodes. There are concerns about scalability of the proposed method and its novelty over the GGP based construction of Caron and Fox. The reviewers also note that the paper needs proof reading and a clearer exposition.

**Award:**

No

---

### Decision · Program_Chairs · 2022-09-14

Accept